# Antifouling graphene oxide membranes for oil-water separation via hydrophobic chain engineering

Chao Yang [1], Mengying Long [1], Cuiting Ding[1], Runnan Zhang [1,2,3], Shiyu Zhang[4], Jinqiu Yuan [1], Keda Zhi[1], Zhuoyu Yin[1], Yu Zheng[1], Yawei Liu [5] ✉, Hong Wu [1,2,3,6] ✉ & Zhongyi Jiang [1,2,3,4] ✉

Engineering surface chemistry to precisely control interfacial interactions is crucial for fabricating superior antifouling coatings and separation membranes. Here, we present a hydrophobic chain engineering strategy to regulate membrane surface at a molecular scale. Hydrophilic phytic acid and hydrophobic perfluorocarboxylic acids are sequentially assembled on a graphene oxide membrane to form an amphiphilic surface. The surface energy is reduced by the introduction of the perfluoroalkyl chains while the surface hydration can be tuned by changing the hydrophobic chain length, thus synergistically optimizing both fouling-resistance and fouling-release properties. It is found that the surface hydration capacity changes nonlinearly as the perfluoroalkyl chain length increases from $C_4$ to $C_{10}$, reaching the highest at $C_6$ as a result of the more uniform water orientation as demonstrated by molecular dynamics simulations. The as-prepared membrane exhibits superior antifouling efficacy (flux decline ratio <10%, flux recovery ratio ~100%) even at high permeance (~620 L m$^{-2}$ h$^{-1}$ bar$^{-1}$) for oil-water separation.

Massive oily wastewater is continuously being generated with the rapid development of industry, especially in petrochemicals, machining, textile, and food industries[1,2]. For example, global produced water production is estimated to be around 250 million barrels per day[3]. Given the discharge standards of most nations for the oil content of effluents (<40 mg/L), there raises a great demand for advanced treatment of oily wastewater[4]. Membrane technology has unique advantages in dealing with oily wastewater for its high separation efficiency and low energy cost, especially for the disposal of the highly stable oil-water emulsion with a droplet size smaller than 20 μm[5,6]. The major obstacle for the membrane separation technique to overcome is the fouling issue. The spreadable oil droplets are susceptible to coalesce

and spread on membrane surfaces, which results in severe membrane fouling, lower separation efficiency, and higher operational costs[7,8]. Two-dimensional graphene oxide (GO)-based membranes show great promise in wastewater treatment for their high flux[9–12]. However, at high flux, the intensified concentration polarization effect will lead to more oil droplets at the surface and induce stronger interaction of the membrane surface with oil droplets[13]. Tailoring membrane surfaces to mitigate the severe fouling at high flux is thus a critical challenge for developing membranes for oil-water separation[14,15].

Minimizing the interfacial interactions of surface to pollutants is the key to enhancing antifouling property[7,16]. A recent promising strategy is constructing amphiphilic surfaces that incorporate

[1]Key Laboratory for Green Chemical Technology, School of Chemical Engineering and Technology, Tianjin University, Tianjin 300072, China. [2]Zhejiang Institute of Tianjin University, Ningbo, Zhejiang 315201, China. [3]Haihe Laboratory of Sustainable Chemical Transformations, Tianjin 300192, China. [4]Joint School of National University of Singapore and Tianjin University, International Campus of Tianjin University, Binhai New City, Fuzhou 350207, China. [5]Beijing Key Laboratory of Ionic Liquids Clean Process, CAS Key Laboratory of Green Process and Engineering, State Key Laboratory of Multiphase Complex Systems Institute of Process Engineering, Chinese Academy of Sciences, Beijing 100190, China. [6]Tianjin Key Laboratory of Membrane Science and Desalination Technology, Tianjin University, Tianjin 300072, China. ✉e-mail: ywliu@ipe.ac.cn; wuhong@tju.edu.cn; zhyjiang@tju.edu.cn

hydrophobic materials (typically fluorinated) onto hydrophilic surfaces to simultaneously realize fouling resistance and release, which has been demonstrated to be effective in numerous fields, including medical devices, marine antifouling, and membrane separation[17–22]. The steric and energetic barrier of hydration layers induced by the hydrophilic domains resists the spread of pollutants[23–26], while the low surface energy of the hydrophobic domains facilitates the release of pollutants[27–29]. Although the amphiphilic antifouling surfaces have been widely reported, achieving the synergic optimization of fouling-resistance and fouling-release properties is still challenging[30]. Generally, nonpolar hydrophobic domains used to enhance fouling-release ability usually reduce the surface hydration, which thermodynamically promotes the spread of pollutants and thus deteriorates the fouling-resistance ability[31,32]. To introduce hydrophobic domains while maintaining the surface hydration capacity, surface chemistry needs to be engineered to elaborately regulate the interfacial interactions between surfaces and pollutants[33]. Previous work attempted to perform the regulation mostly by adjusting the hydrophobicity, amount, and distribution of hydrophobic domains[33–36]. So far, few studies have achieved the optimization of the interfacial interaction by interfering with the surface hydration of the amphiphilic surfaces. Additionally,

the underlying molecular-level influence of hydrophobic domains on the hydration capacity of amphiphilic surfaces is still elusive.

Herein, we constructed an amphiphilic GO membrane and proposed a hydrophobic chain engineering strategy to regulate the interfacial interactions of the surface with oil droplets, thus achieving ultralow fouling at high flux. We sequentially assembled hydrophilic phytic acid (PA) and hydrophobic perfluorocarboxylic acids on a GO membrane to form an amphiphilic surface with a continuous hydrophilic domain and discrete hydrophobic domains. For the construction of PA on membrane surfaces, assembly strategies based on hydrogen bonds and coordination are recently proposed[37–39]. Coordination-driven metal-bridging assembly, in our study, is employed to perform the controllable assembly of PA and perfluoroalkyl chains. The surface energy is reduced by introducing the perfluoroalkyl chains, while concomitantly, the surface hydration can be tuned by changing the length of the hydrophobic chains, thus achieving the synergetic enhancement in both fouling-release and fouling-resistance properties. We systematically analyzed the wetting behavior of the proposed membrane to illustrate the effect of the length of perfluoroalkyl chains on interfacial interactions and correlated them with antifouling properties. Further, molecular dynamics simulation, together with experi-

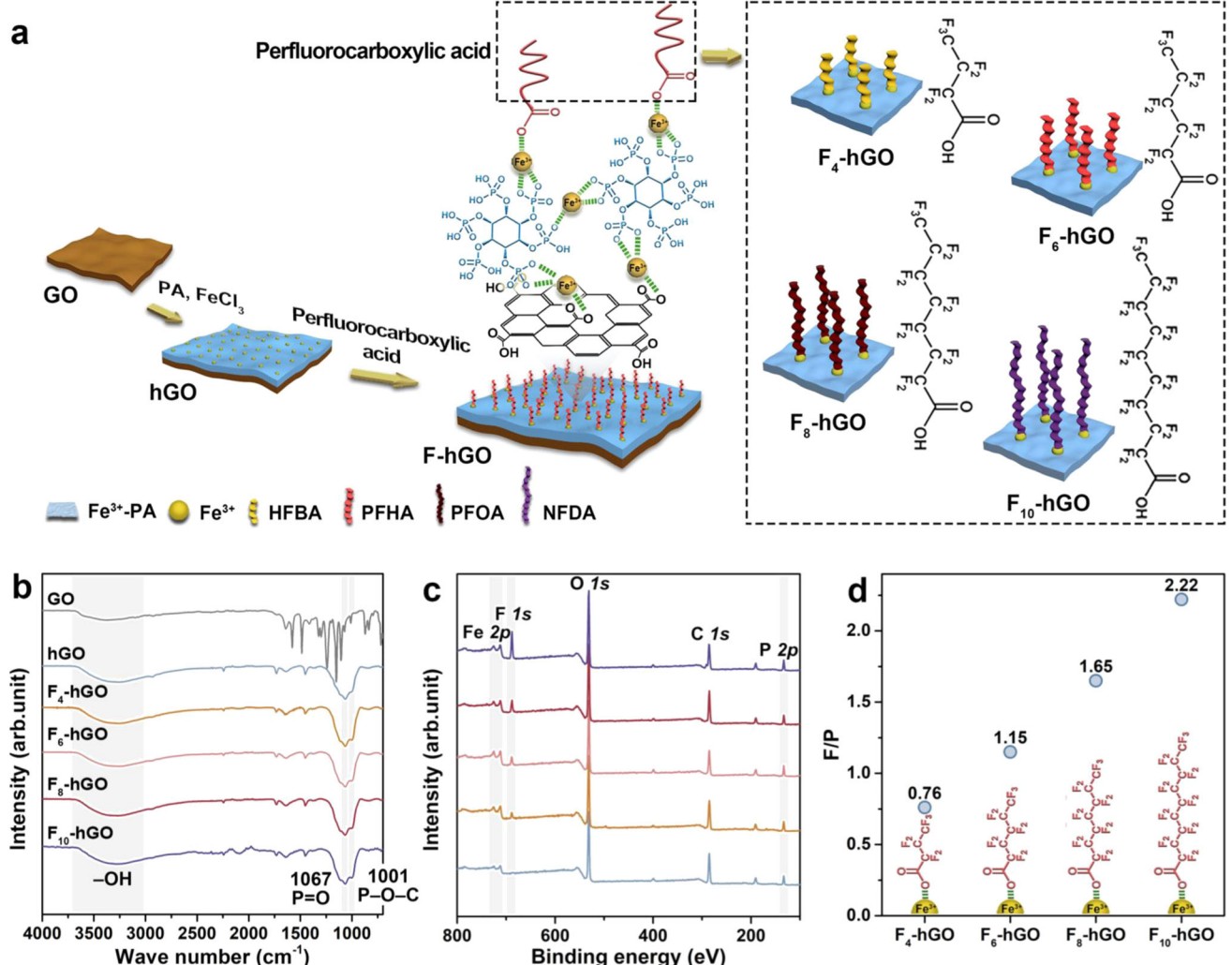

**Fig. 1 | Preparation and structures of the membranes. a** Schematic diagram of the fabrication of the F-hGO membranes. HFBA, PFHA, PFOA, and NFDA are heptafluorobutyric acid, perfluorohexanoic acid, pentadecafluorooctanoic acid, and nonadecafluorodecanoic acid, respectively. **b** FTIR spectra of the GO, hGO, and F-hGO membranes. **c** XPS spectra of the hGO and F-hGO membranes. **d** The calculated F to P ratio of the F-hGO membranes determined by the peak area on XPS spectra. Source data are provided as a Source Data file.

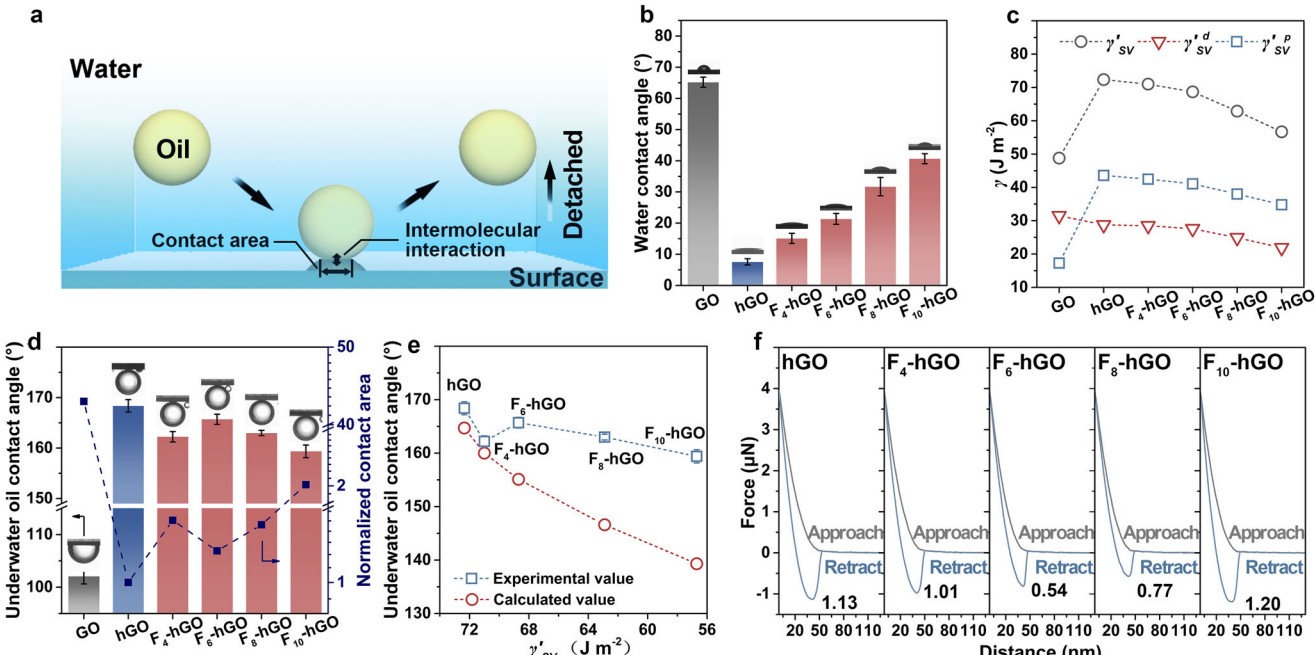

**Fig. 2 | Wetting behavior of the membranes. a** Schematic diagram of the removal of oil droplets from a solid surface. **b** Water contact angles, **c** apparent surface energy ($\gamma'_{SV}$), **d** underwater oil contact angles and normalized contact areas of the GO, hGO, and F-hGO membranes. The areas are normalized by the contact area of the hGO membrane. **e** Dependence of the underwater oil contact angle of the membranes on the apparent surface energy. **f** Underwater oil adhesive force on the hGO and F-hGO membranes. The oil is hexadecane. All error bars in this figure represent standard deviations for three measurements. Source data are provided as a Source Data file.

ments, were employed to elucidate the evolution of the hydration structure on the proposed amphiphilic membrane. Finally, we applied the membrane in the filtration of oily wastewater and demonstrated the enhancement in the antifouling capacity during the separation process.

## Results

### Regulation of the surface chemical structure of the F-hGO membranes

PA possessing six phosphate groups with high hydration energy and perfluorocarboxylic acid with ultralow surface energy were selected as the hydrophilic and hydrophobic components, respectively, and the amphiphilic surfaces were prepared on GO membranes via coordination-driven metal-bridging assembly. We chose per-fluorocarboxylic acids with different numbers of carbon atoms, namely, 4, 6, 8, and 10, to adjust the length of the hydrophobic chains and changed the concentration of $Fe^{3+}$ on the surface to control the density of the perfluorocarboxylic acids, hence achieving the molecular-scale regulation of the surface chemistry. Specifically, as described in Fig. 1, during the fabrication, the $Fe^{3+}$-PA complexes grew on GO nanosheets via coordination, and a superhydrophilic GO membrane (hGO) was formed by vacuum-assisted self-assembly. The presence of the characteristic peaks of phosphate bond (1001 and 1067 cm$^{-1}$) in the Fourier transform infrared (FTIR) spectrum of the hGO membrane (Fig. 1b), together with the appearance of the phosphate peaks (132.7 and 133.6 eV) in the X-ray photoelectron spectroscopy (XPS) spectrum of the hGO membrane (Fig. 1c and Supplementary Fig. 3a), illustrates the introduction of $Fe^{3+}$–PA complex on the GO membrane[37,38]. $Fe^{3+}$ acts as bridges and incorporates the hGO membrane surface with the carboxylic acid of the perfluorocarboxylic acids, thus anchoring the perfluorocarboxylic acids and forming the amphiphilic surfaces. The C–F bond can be detected in the XPS spectra (688.2 eV, Fig. 2c and Supplementary Fig. 3b) of the F-hGO membranes, which demonstrates the incorporation of perfluoroalkyl chains on the hGO membrane[40].

The variation in the length of perfluorocarboxylic acids is supposed to exert a negligible effect on the coordination process, which can be proved quantitatively by the area ratio of the F peak to the P peak in the XPS spectra of the F-hGO membranes. The peak areas of the P element are almost consistent in the four cases (Supplementary Fig. 3a), suggesting the detection of the P element is not significantly affected by the chain length, which illustrates the validity of the F/P ratio to characterize the F content. As shown in Fig. 1d, the F content continuously ascends with the chain length, and its variation is almost in line with the molar ratio of the F atoms in the introduced perfluorocarboxylic acids, implying the distribution density of the coordinated perfluorocarboxylic acids on the four surfaces is almost similar. Further, we observed the $F_6$-hGO membrane without substrate under transmission electron microscopy. The energy-dispersive X-ray spectroscopy mapping image reveals that the perfluorocarboxylic acids are evenly distributed on the membrane surface (Supplementary Fig. 4).

### Wetting behavior

The wetting behavior of the membranes was studied to assess the effect of the length of perfluoroalkyl chains on the interfacial interactions. Generally, a low intermolecular interaction and a small contact area lead to low adhesion and thus help the removal of oil droplets (Fig. 2a)[27,41]. Modulating the length of perfluoroalkyl chains, we observed that the apparent water contact angle monotonously increases with the chain length (Fig. 2b). The decline in apparent surface energy ($\gamma'_{SV}$) caused by the increased F content accounts for the increase in water contact angle (Fig. 2c). The decreasing $\gamma'_{SV}$, especially the dispersion component ($\gamma'^d_{SV}$), indicates that the Van der Waals interaction between the surface and the environment descends with the prolongation of hydrophobic chains. On the other hand, the reduced surface energy thermodynamically promotes the spread of oil droplets on surfaces[42]. In our case, the underwater oil contact angle reduces, but it still exhibits underwater superoleophobicity (Fig. 2d). It is worth noting that the underwater oil contact angle is not linearly

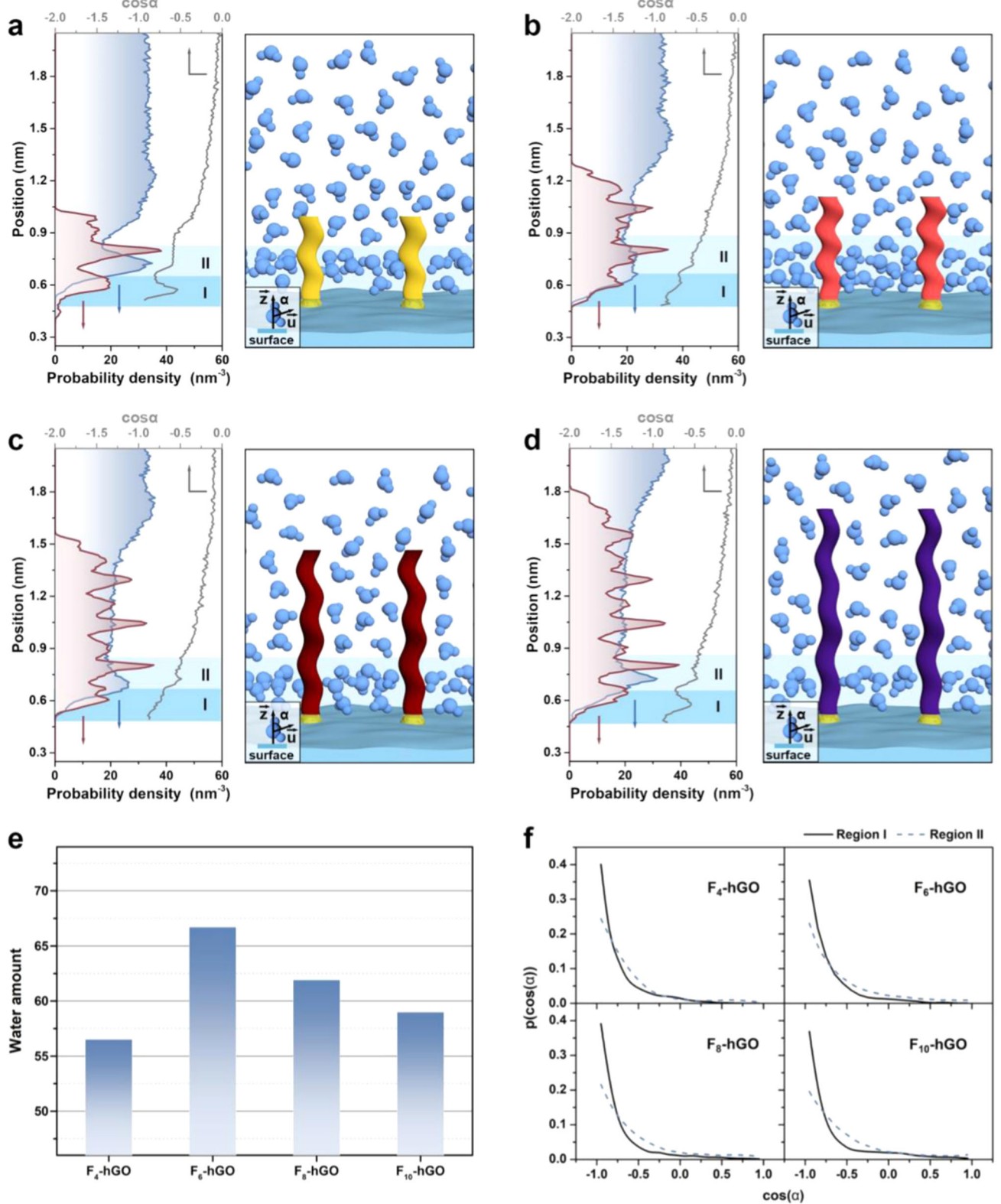

**Fig. 3 | Hydration structures on the surfaces. a–d** Water density (blue line), F density (red line), and cosα (gray line) along the direction perpendicular to the proposed surfaces and the corresponding schematic diagram of the hydration structure of the proposed surfaces. α is defined as the angle between **ū** (the opposite vector of the water dipole) and **z̄** (the normal vector of the surface).

**e** Water amount in the hydration layer of the proposed surfaces. **f** Probability distributions of the orientation of the water in the hydration layer of the proposed surfaces. Solid and dashed lines are the distributions for regions I and II, respectively. Source data are provided as a Source Data file.

related to the chain length but show an inverted-U tendency. The $F_6$-hGO membrane displays the highest underwater oil contact angle (165.7°) and smallest contact area compared to those of other F-hGO membranes. We further analyzed the relationship between the $\gamma'_{SV}$ and the underwater oil contact angle and calculated the corresponding theoretical values according to Young's equation in solid−liquid−liquid systems and Wenzel's equation[43,44].

$$\cos\theta_{OW} = \frac{\gamma_{SW} - \gamma_{SO}}{\gamma_{OW}} \qquad (1)$$

$$\cos\theta'_{OW} = r\cos\theta_{OW} \qquad (2)$$

where $\theta_{OW}$ and $\theta'_{OW}$ are the underwater oil contact angles on the smooth and rough surfaces, respectively. $\gamma_{SW}$, $\gamma_{SO}$, and $\gamma_{OW}$, are the interfacial tension of the surface/water, surface/ hexadecane, and hexadecane/water, respectively. $r$ is the roughness factor of the surfaces, which was obtained from the AFM test (Supplementary Fig. 5). The result is presented in Fig. 2e and Supplementary Table 1. We noticed that the experimental values are all higher than the calculated values. This deviation can be observed in many works, which is mainly because surface hydration changes the interfacial interactions of the system[45,46]. More importantly, we find that the curve shape deviates from the calculated one in which the underwater oil contact angle continuously decreases with the surface energy. The $F_6$-hGO membrane with lower surface energy shows higher underwater oleophobicity compared to that of the $F_4$-hGO membrane. This unexpected result suggests that the surface hydration is sensitive to the chain length, which allows us to control the surface hydration when introducing perfluoroalkyl chains to simultaneously achieve both a low contact angle and a low intermolecular interaction. To illustrate the universality of the phenomenon, we also tested the underwater oil contact angles with tetrachloromethane and n-hexane, and similar results were obtained (Supplementary Fig. 6).

Adhesive force, as shown in Fig. 2f, was further measured by AFM to quantitatively probe the oil-membrane interaction between the amphiphilic membranes and oil droplets. As expected, due to the lower surface energy and the smaller oil contact area, the adhesive force of the $F_6$-hGO membrane (0.54 μN) drops nearly 50% compared with that of the $F_4$-hGO membrane (1.01 μN). When the number of carbon atoms in the perfluoroalkyl chains exceeds 6, extending the perfluoroalkyl chains can further reduce the surface energy, but the excessively increased contact area substantially rises interfacial interactions. For instance, in terms of the $F_{10}$-hGO membrane, the surface energy decreased by -15%, while the contact area increased by more than 60% compared with those of the $F_6$-hGO membrane, eventually leading to a 1.20 μN adhesive force.

## Hydration capacity

Our experiments presented an unexpected phenomenon that the underwater oil contact angle of the proposed membrane surface shows a nonmonotonic relationship with the length of perfluoroalkyl chains. This endows the amphiphilic membrane with the medium-length perfluoroalkyl chains ($C_6$) with the lowest contact area, thereby obtaining the lowest underwater oil adhesive force. Surface hydration has been proven to be associated with underwater oil-wetting behaviors[47]. More and more studies pointed out that the side groups or functional groups within the size range of the hydration layers (subnanometer to few nanometers) can control the hydration structure, hence affecting interfacial behavior and antifouling performances[24,48]. Inspired by these, we suppose that perfluoroalkyl chains with different lengths alter the hydration structure of the surface, hence regulating the distribution density and cohesive strength of the water adsorbed on surfaces.

To confirm our hypothesis, molecular dynamics simulations were performed to elucidate the evolution of the hydration on the proposed amphiphilic surfaces. The perfluoroalkyl chains, due to their nonpolar nature, is not prone to attach to the polar superhydrophilic surfaces in the water[49]. Therefore, the perfluoroalkyl chains were set as stretching outward to the aqueous phase in our model. Figure 3a−d display the plots of the water density along the direction perpendicular to the surfaces. Due to the coulombic interaction between the superhydrophilic substrate and water molecules[50], an interfacial water layer (<-0.6 nm in our case) with a density higher than that of the bulk water is formed on all the surfaces. It can be found that the sharpness of the peak of the interfacial water layer first decreases and then increases with the length of the perfluoroalkyl chain, suggesting the structure of the hydration layer displays a varying tendency of first decreasing and then increasing. The hydration layer induced on the $F_6$-hGO surface possesses wider and more uniform density distribution. Some studies proved that moderately interfering with the hydration layer can increase the amount of interfacial water, thus achieving a stronger hydration capacity[51]. We also observed a change in the water amount in the hydration layer, that is, the value first increases and then decreases with the perfluoroalkyl chain length (Fig. 3e). The variation in hydration capacity is considered to be related to the interfacial hydration structure induced by the perfluoroalkyl chains with different lengths, which will be discussed in detail later. Further, differential scanning calorimetry was used to monitor the crystallization temperature and crystallization enthalpy of water (-10 mg) on the membrane surfaces to evaluate the amount of interfacial water (Supplementary Figs. 7, 8). All the two values follow the order of $F_4$-hGO > $F_8$-hGO > $F_6$-hGO, illustrating the amount of interfacial water in the order of $F_6$-hGO > $F_8$-hGO > $F_4$-hGO, which is well consistent with the simulation results. It is believed that more water pinning on surfaces provides a stronger steric and energetic barrier[24]. These results explain the unexpected nonlinear relationship between the perfluoroalkyl chain length and the underwater oil contact angle.

We defined α as the angle between $\vec{u}$ (the opposite vector of the water dipole) and $\vec{z}$ (the normal vector of the surface) to estimate the spatial orientation of interfacial water to explore the underlying molecular-level details for the variation in the water amount of the hydration layer. We divide the hydration layers into two regions and compare the probability distributions of cosα in the two regions (Fig. 3f). It can be seen that most of the water bears the distributions of cosα centered -−1, implying water mainly favor orienting with two O−H bonds directed downwards towards the surface in our models. With the presence of perfluoroalkyl chains, the orientation of the water molecules is perturbed due to the electrostatically dominated interactions between the water and the perfluoroalkyl chains[52]. This electrostatic interaction is sensitive to chain length and position, which was adequately discussed in detail in the previous work[52]. In our study, the water orientation distribution in regions I and II is more similar in the case of $F_6$-hGO, which implies a more uniform hydration structure on the surface. The uniformity in hydration structure offers more available water interaction sites and thus reaches a larger amount of interfacial water and higher hydration capacity[50]. This nonlinear dependence of the hydration capacity to the perfluoroalkyl chain length reveals that an appropriate chain length exists for similar amphiphilic surfaces to exhibit the strongest hydration capacity.

## Antifouling efficacy

Based on the regulation of interfacial interactions by the length of perfluoroalkyl chains, we tried to explore its effect on the antifouling efficacy. Hexadecane-in-water emulsion (average size = 291.1 nm, Supplementary Fig. 9) was selected as the target separation system to evaluate antifouling performances. All membranes bear high oil repellency (Rejection >98%, Supplementary Fig. 10) at high permeance (-620 L m$^{-2}$ h$^{-1}$ bar$^{-1}$). The oil concentration of the filtrates is less than

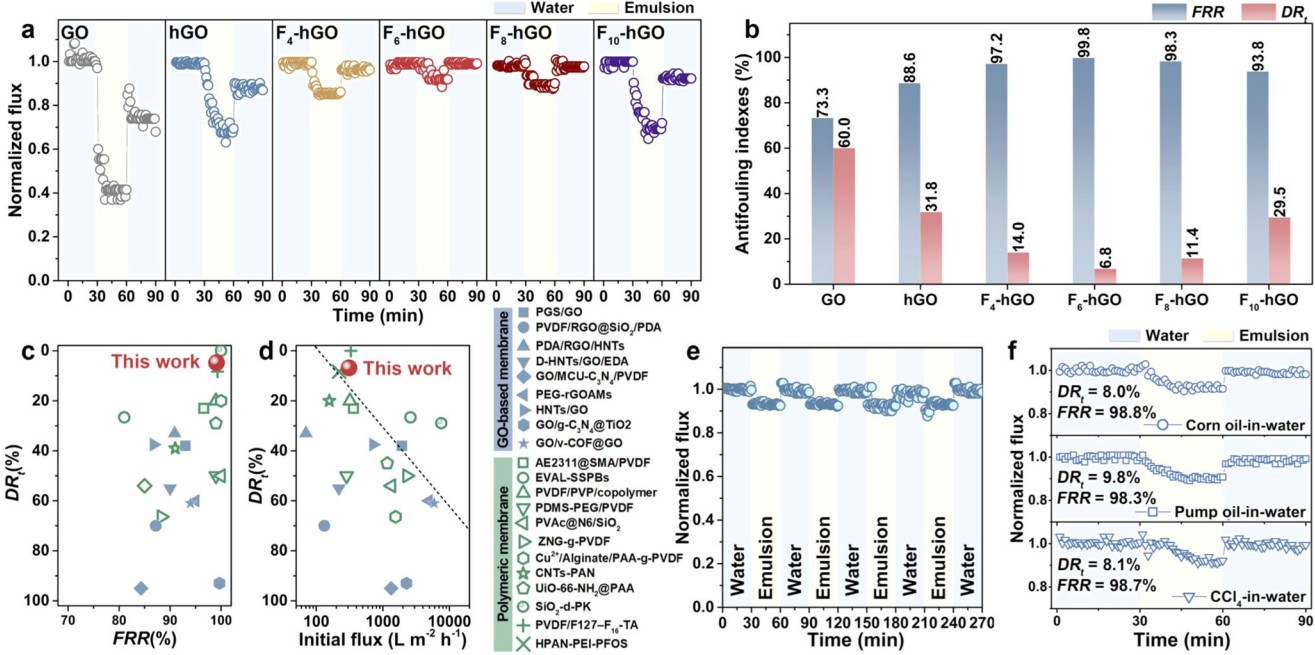

**Fig. 4 | Antifouling performances of the membranes. a** Time-dependent normalized fluxes of the GO, hGO, and F-hGO membranes during antifouling tests. The used oily wastewater is a hexadecane-in-water emulsion. **b** Antifouling indexes of the GO, hGO, and F-hGO membranes. **c** Comparison of antifouling performances of the $F_6$-hGO membrane and the reported membranes for oil-water separation. **d** $DR_t$ and initial flux of the $F_6$-hGO and the reported membranes for oil-water separation. **e** Cycling performances of the $F_6$-hGO membrane for the separation of hexadecane-in-water emulsion. **f** Normalized flux of the $F_6$-hGO membrane varied with the processing time for the separation of different oil-in-water emulsions. Source data are provided as a Source Data file.

20 ppm (Supplementary Fig. 11), which meets the discharge standard of most nations[4]. The fluxes varying with time during filtration and the corresponding indices are illustrated in Fig. 4a, b. The antifouling capability quantized by $DR_t$ first decreases and then increases with the chain length, which is consistent with variation in adhesive force (Fig. 2f). It can be significantly reduced by more than 50% by the hydrophobic chain engineering. Additionally, we investigated the effect of the density of the perfluoroalkyl chains on the antifouling properties to explore the optimal chemical composition of the proposed amphiphilic membrane surface (Supplementary Fig. 12). Our results indicate that the amphiphilic membrane with medium-length perfluoroalkyl chains ($C_6$) prepared at the immersed $FeCl_3$ solution concentration of 0.6 mM possesses the lowest $DR_t$ (6.8%) and the highest $FRR$ (99.8%). Its $DR_t$ is dramatically reduced by nearly 90 and 80%, respectively, and its FRR is improved by 1.4 and 1.1 times, respectively, relative to those of the GO membrane ($DR_t = 60.0$ %, FRR = 73.3%) and the hGO membrane ($DR_t = 31.8$ %, FRR = 88.6%).

Compared with the currently reported GO-based membranes and polymeric membranes for oil-water separation, the $F_6$-hGO membrane shows superior antifouling performance (Fig. 4c, d and Supplementary Tables 2, 3). $DR_t$ of our membrane is far lower than that of the reported membranes, with the index generally of 40–60%. More importantly, as shown in Fig. 4d, most membranes achieve ultralow $DR_t$ at low initial fluxes. While the $F_6$-hGO membrane can maintain its ultralow fouling tendency even at a relatively high initial flux of ~310 L m$^{-2}$ h$^{-1}$. This value exceeds most oil-water separation membranes, which confirms their robustness in resisting oil fouling and their advantage over state-of-the-art membranes. In addition, the membrane exhibits a comparable pure water flux as the initial one after multiple cycles of fouling with oils and cleaning with water (Fig. 4e). For the challenge of scaling, no significant flux decline was observed after operating the gypsum scaling experiment for 6 h (Supplementary Fig. 13). Besides, the $F_6$-hGO membrane shows a promising opportunity in the antifouling treatment of various

emulsions with high rejection (>98%, Supplementary Fig. 14) and low $DR_t$ (<10%, Fig. 4f).

The introduction and optimization of multiple antifouling mechanisms on the membrane surface account for excellent antifouling properties. The hydration barrier and steric repulsion of the hydrophilic PA coating resist the spread of oil droplets, making the oil contact the membrane surface with a small area[53]. The perfluoroalkyl chains with low surface energy endow the membrane surface with nonstick properties and fouling-release behavior[27]. Most importantly, in our case, the interfacial interactions of the proposed amphiphilic surface are regulated by the perfluoroalkyl chain length. There is an optimal chain length, namely $C_6$, to endow the surfaces with the lowest oil adhesion. At this length, the interfacial water is oriented more uniformly, thereby forming a hydration layer with more water. This enables the surface to maintain the hydration capacity when introducing the perfluoroalkyl chains, thereby synergistically optimizing fouling resistance and release properties. Consequently, the proposed $F_6$-hGO membrane harvests ultralow adhesion with oil droplets, allowing oil droplets to easily peel from the membrane surface instead of forming a filter cake layer and thus resulting in enhanced antifouling efficacy.

## Discussion

In summary, we constructed an amphiphilic surface by sequentially assembling hydrophilic PA and hydrophobic perfluoroalkyl chains on a graphene oxide membrane and achieved molecular-scale regulation of its chemical structure and interfacial interactions by adjusting the density and length of the perfluoroalkyl chains. Our study demonstrates that the perfluoroalkyl chain length is one of the key factors affecting the antifouling ability, because it can tune the surface hydration structure, thus synergistically optimizing both fouling-resistance and fouling-release properties. We observed that the hydration capacity varies nonlinearly with the perfluoroalkyl chain length and proved that the medium-length perfluoroalkyl chains ($C_6$) is

optimal by providing a more uniform orientation of the interfacial water to hydrate more water on the surface. This hydrophobic chain engineering reduces the adhesive force of oil droplets by over 50%, thus achieving superior antifouling capability ($DR_t$: <10%, FRR: ~100%) at even high permeance (~620 L m$^{-2}$ h$^{-1}$ bar$^{-1}$). Our work provides guidance for the regulation of surface interactions. We anticipate that the microscopic understanding of interfacial interaction is also instructive to interfacial engineering in oil recovery, biomedicine, and nanofluidics.

## Methods

### Materials and chemicals
GO nanosheets were purchased from Nanjing XFNANO Materials Tech Co., Ltd. Polyacrylonitrile (PAN) membranes were provided by Lanjing Membrane Engineering Co. PA, iron chloride hexahydrate (FeCl$_3$·6H$_2$O), sodium dodecyl sulfonate, tetrachloromethane, n-hexane, and hexadecane were supplied from Tianjin heowns Biochemical Technology Co., Ltd. HFBA, PFHA, PFOA, NFDA, and diiodomethane were obtained from Shanghai Macklin Biochemical Co., Ltd. Vacuum pump oil was provided by Special Oil Factory. Corn oil was bought from COFCO Fulinmen Co., Ltd.

### Preparation of the hGO membrane
Fe$^{3+}$−PA complex was grown on GO nanosheets via the metal-organic coordination[37]. Typically, as described in Fig. 1a, 0.50 mg GO nanosheets and 49.50 mg PA were uniformly dispersed in 25 mL deionized water. After shaking at 150 rpm and 25 °C for 10 min, 5 mL 0.20 mM FeCl$_3$ aqueous solution was added into the mixed solution of GO and PA and then shaken for 10 min to fabricate Fe$^{3+}$−PA@GO nanosheets. Subsequently, 1800 μL of the above-resulting Fe$^{3+}$−PA@GO nanosheet solution was ultrasonically dispersed in 100 mL of water. The hGO membrane was prepared via filtrating the Fe$^{3+}$−PA@GO dispersion onto the PAN membranes.

### Preparation of the F-hGO membranes
F-hGO membranes were fabricated by sequentially assembling Fe$^{3+}$ and perfluorocarboxylic acids on the hGO membrane[38]. As shown in Fig. 1a, the hGO membrane was first immersed into 30 mL 0.60 mM FeCl$_3$ aqueous solution and shaken for 10 min at 100 rpm and 25 °C, followed by washing with water. Afterward, the resulting membranes were sunk into the 30 mL ethanol solution of perfluorocarboxylic acids and shaken at 100 rpm and 25 °C for 10 min. The F-hGO membranes were obtained after washing with ethanol. The subscript of the membrane name indicates the number of C atoms in the molecular chain of perfluorocarboxylic acids, that is, F$_4$-hGO, F$_6$-hGO, F$_8$-hGO, and F$_{10}$-hGO represent the membranes assembled by HFBA, PFHA, PFOA, and NFDA, respectively. The concentration of the immersed FeCl$_3$ solution, namely, 0.20, 0.40, 0.60, and 0.80 mM, was adjusted to control the density of the perfluorocarboxylic acids. The concentration of perfluorocarboxylic acids solution was set to an excessive concentration (2.00 mM) to achieve the adequate coordination of perfluorocarboxylic acids with Fe$^{3+}$, according to the influence of the concentration of perfluorocarboxylic acids on the surface properties quantified by the water contact angle (Supplementary Fig. 1).

### Molecular dynamics simulation
Molecular dynamics simulations with a simple model system were used to investigate the interfacial water on F-hGO membranes with perfluoroalkyl chains of different lengths. Supplementary Fig. 2 shows a typical box used in our simulations. There are 1000 water molecules on the substrate. The substrate is made up of a group of randomly arranged atoms with a density of 0.005 nm$^{-3}$ (i.e., there are 259 substrate atoms in the simulation box). The substrate is negatively charged with the total charges Q = −10 e where e is the charge of an electron (we checked that this type of substrate is superhydrophilic).

Na$^+$ ions are added in the system to neutralize the system. 12 perfluorocarboxylic acid chains are anchored on the substrate with the C atom of the −COOH groups fixed at the top of the surface (i.e., at z = 0.5 nm in Supplementary Fig. 2). The perfluorocarboxylic acids are arranged in a hexagonal lattice with the distance between them d = 1 nm. The cases anchored by the perfluorocarboxylic acids of HFBA, PFHA, PFOA, and NFDA are named F$_4$-hGO, F$_6$-hGO, F$_8$-hGO, and F$_{10}$-hGO, respectively.

The SPC/E model was applied to the water molecules[54]. The SHAKE algorithm was used to maintain the rigidity of the water. The force field for the perfluoroalkyl chains were generated by LigParGen[55]. The model for Na$^+$ ions was from ref. 56. The substrate atom was set to be the same as the O atom of H$_2$O (except for the charges). Non-bonding pairwise interaction was given by

$$U = 4\epsilon_{ij}\left[\left(\frac{\sigma_{ij}}{r_{ij}}\right)^{12} - \left(\frac{\sigma_{ij}}{r_{ij}}\right)^{6}\right] + \frac{q_i q_j}{4\pi\epsilon_0 r_{ij}} \qquad (3)$$

where $\epsilon_{ij}$ and $\sigma_{ij}$ are the energy and distance parameters, respectively, $r_{ij}$ is the distance between atoms $i$ and $j$, $q_i$ and $q_j$ are the atomic charge of atoms $i$ and $j$, $\epsilon_0$ is the vacuum permittivity. The Lennard-Jones (LJ) potentials were cut and shifted at 1.0 nm. The arithmetic mixing rule was applied for LJ potentials between different species. The electrostatic interaction was also cut at 1.0 nm and the particle−particle particle−mesh (PPPM) with an accuracy of 10$^{-4}$ was employed to calculate the long-range electrostatic interactions.

All simulations were carried out using the parallel molecular dynamics software package LAMMPS[57]. Periodic boundary conditions were imposed in all three directions. The velocity-Verlet algorithm with a time step of 1 fs was used to integrate the equation of motion, and a Nosé−Hoover thermostat with a time constant of 100 fs was used to maintain the temperature of the fluid at T = 300 K. During all simulations, all atoms of the substrates and the C atom of the −COOH groups were frozen at their initial positions. Each simulation was run for 2 ns for equilibrating and another 10 ns for producing.

### Characterization
Characterization used in this work are provided in detail in the Supplementary Methods.

## Data availability
All data supporting the findings of this study are available within the article and the Supplementary Information file, or available from the corresponding authors upon request. Source data are provided with this paper.

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

## Acknowledgements

This work is financially supported by the National Natural Science Foundation of China, Grant No. 21961142013 (Z.J.), 21878215 (H.W.), Key Research and Development Program of Zhejiang Province, Grant No. 2021C03173 (Z.J.), and the Haihe Laboratory of Sustainable Chemical Transformations for financial supports. The authors also appreciate Dr. Mingrui He from the Harbin Institute of Technology for his help with the AFM test. Y.L. sincerely appreciates Prof. Suojiang Zhang (IPE, CAS) for his careful aca- demic guidance and great support.

## Author contributions

H.W., Z.J., and C.Y. conceived the idea and designed the experiment. C.Y., M.L., S.Z., J.Y., and C.D. performed the experiment. K.Z., Z.Y., and Y.Z. measured the separation performances. R.Z provided constructive suggestions for results and discussion. Y.L. did molecular dynamics simulations. All authors participated in the discussion.

## Competing interests

The authors declare no competing interests.
