## [Peer Review File · Nature Communications]

Antifouling graphene oxide membranes for oil-water separation via hydrophobic chain engineeringREVIEWER COMMENTS

Reviewer #1 (Remarks to the Author):

The authors investigated the effect of hydrophilic PA and hydrophobic perfluoroalkyl chain length on the graphene oxide as a membrane on antifouling efficacy. The chain length affected interfacial interactions. The constructed membranes will be used for oil separation in the wastewater. The manuscript is well written and presents findings very clearly and well organized. Huge thanks to the authors. I believe that the manuscript can be published in its current form.

Best regards

Reviewer #2 (Remarks to the Author):

Supplementary

Under subsection 1 Experimental, 1.3 preparation of the F-hGO provided the approach used to obtain the number of C atoms in the molecular chain of perfluorocarboxylic acids, authors should add the variation in the concentrations of perfluorocarboxylic acids to cover F6-hGO, F8-hGO, and F10-hGO.

Subsection 1.5 separation evaluation, how did you attain emulsified oil stabilization considering the mixture of two immiscible liquids, and what is the role of SDS?

Also, in subsection 1.5 antifouling evaluation, what informed the choice of 0.5 bar to test the antifouling performance and what types of membrane processes were constructed for this study?

Main File

Figure S4 AFM showed a different variation in the r - values despite an increment and the sequential assembling of Fe^{3+} and perfluorocarboxylic acid concentrations. It was expected that the roughness factor should increase with the loading of perfluorocarboxylic acids. However, Line 115 page 6 further suggested that perfluorocarboxylic acid possesses less significance in the chain length which resulted in amphiphilic surfaces. Authors should recheck the order of roughness factor since they were the basis for obtaining Table S1.

Authors should explain further in addition to Line 208-210 page 11, why F6-hGO possesses the highest hydration capacity compared to other membranes as depicted in Figure 3(e) F6-water amount in the hydration layer of the proposed surfaces?

Authors should place all figures within or next to the text of the subsection, not above the text or paragraph.

Line 284 page 14, authors should be presented as "3 Conclusion" instead of "3 Discussion".

Reviewer #3 (Remarks to the Author):

This is an interesting article about self-assembled amphiphilic structures on membranes. While some properties like the contact angle rise as expected with increasing alkyl chain length, other properties like the oil adhesion force show a more differentiated picture. Obviously, hydration and morphology are more relevant than pure wettability. The work is well backed up by state-of-the-art spectroscopy. The interpretation of the data is supported by simulations – but I am unable to

judge their quality. What is missing is the question how the coatings deal with scaling – I recommend additional experiments to add this characterization. I find the publication interesting, and I am in favor of its publication after major revision.

Detailed comments

In the introduction on amphiphilic coatings as marine antifouling coatings, the list of references seems incomplete and the important contributions of groups like the Ober group (Cornell), the Webster group (NDSU), and the Laschewsky group (Potsdam) are not mentioned. This should be extended. Also, the work of the Ulbricht group (University of Duisburg-Essen) about amphiphilic coatings in membrane science is not cited. This overview needs to be complete.

In the introduction, the state of the art for the used assembly strategy using PA and PFCA on GO is missing and needs to be added.

While the selection of perfluorocarboxylic acid is discussed in detail, the choice of phytic acid is not explained. The authors should add why the specific compound has been selected.

In the XPS analysis on page 6, line 115, the F/P ration was determined to understand the stoichiometry of the two compounds on the surface. As the assembly process is a subsequent buildup on the surface, I am wondering if the F is predominantly present at the surface and the P might be buried? The authors have to briefly argue in the article if the quantification would also be valid if there is a layered system and which deviations could occur, accounting for the high surface sensitivity of the used method.

The discussion section is mainly a summary – here an extended discussion based on the literature state of the art is required with the corresponding references. Compared to which systems is the developed system superior and how does it compare to the current gold standard?

In the materials and methods part in the supplements, literature references are missing. While the theory part is well referenced, for the preparation of the coatings and for the separation and antifouling performance evaluation such reference to previous work is entirely missing.

Response to reviewer's comments

Reviewer #1 (Remarks to the Author):

The authors investigated the effect of hydrophilic PA and hydrophobic perfluoroalkyl chain length on the graphene oxide as a membrane on antifouling efficacy. The chain length affected interfacial interactions. The constructed membranes will be used for oil separation in the wastewater.

The manuscript is well written and presents findings very clearly and well organized. Huge thanks to the authors. I believe that the manuscript can be published in its current form.

Best regards

Thank the reviewer for the highly positive remarks.

Reviewer #2 (Remarks to the Author):

Supplementary

Under subsection 1 Experimental, 1.3 preparation of the F-hGO provided the approach used to obtain the number of C atoms in the molecular chain of perfluorocarboxylic acids, authors should add the variation in the concentrations of perfluorocarboxylic acids to cover F6-hGO, F8-hGO, and F10-hGO.

Reply:

Thank the reviewer for the valuable guidance.

The density of perfluoroalkyl chains mainly depends on two factors, namely, the concentration of Fe^{3+} on the surface and the concentration of perfluorocarboxylic acids in the assembly process. The following figure shows the influence of the concentration of perfluorocarboxylic acids on the surface properties quantified by the water contact angle. As shown in Figure S1, take the F₆-hGO membrane as an example, when the concentration of perfluorocarboxylic acids exceeds 0.5 mM, the water contact angle kept almost constant, suggesting that the coverage of perfluoroalkyl chains on the surface has been maximized. We chose the excessive concentration (2 mM) to prepare the amphiphilic surfaces in order that the only control variable of the surface chemical composition is Fe^{3+} concentration. We have added the above description in the revised version (Supplementary Figure S1 and Manuscript section 4.2):

“The concentration of the immersed FeCl_3 solution, namely, 0.2, 0.4, 0.6, and 0.8 mM, was adjusted to control the density of the perfluorocarboxylic acids. The concentration of perfluorocarboxylic acids solution was set to an excessive concentration (2 mM) to achieve the adequate coordination of perfluorocarboxylic acids with Fe^{3+} , according to the influence of the concentration of perfluorocarboxylic acids on the surface properties quantified by the water contact angle (Supplementary Figure S1).”

Figure S1 Water contact angle of the F-hGO membrane varied with the concentration of the perfluorocarboxylic acids solution.

Subsection 1.5 separation evaluation, how did you attain emulsified oil stabilization considering the mixture of two immiscible liquids, and what is the role of SDS?

Reply:

SDS, as a typical surfactant, emulsifies oil to form stable emulsions. SDS, with an alkyl chain and $-\text{SO}_3\text{Na}$, interacts with oil and water, respectively, thus attaining the stabilization of emulsified oil.

Also, in subsection 1.5 antifouling evaluation, what informed the choice of 0.5 bar to test the antifouling performance and what types of membrane processes were constructed for this study?

Reply:

The type of our membrane process is ultrafiltration on the basis of the membrane pore size. In the current studies of the membrane process for oil/water separation, the commonly used transmembrane pressure is 0.2~1 bar. Low operating pressure is conducive to reducing energy consumption. We chose 0.5 bar as our operating pressure, and under this pressure, our membrane shows competitive separation efficiency,

including the permeation flux and rejection.

Main File

Figure S4 AFM showed a different variation in the r – values despite an increment and the sequential assembling of Fe^{3+} and perfluorocarboxylic acid concentrations. It was expected that the roughness factor should increase with the loading of perfluorocarboxylic acids. However, Line 115 page 6 further suggested that perfluorocarboxylic acid possesses less significance in the chain length which resulted in amphiphilic surfaces. Authors should recheck the order of roughness factor since they were the basis for obtaining Table S1.

Reply:

Thank the reviewer for the valuable guidance. In this work, the variation in the length of perfluorocarboxylic acids is supposed to exert a negligible effect on the coordination process, which can be proved quantitatively by the ratio of F/P that ascends with the molar ratio of the F atoms in the introduced perfluorocarboxylic acids. For surface roughness, regulating the length of perfluorocarboxylic acids does change the surface roughness, but the change is incredibly small because the change in the length of perfluorocarboxylic acids is less than 1 nm. AFM indicates that the roughness factor only changes within the range of ± 0.02 , which would not significantly affect the wetting behavior of the membranes according to the Wenzel equation ($\cos\theta'_{ow} = r\cos\theta_{ow}$). So, in Table S1, the average roughness factor is employed to calculate the $\cos\theta'_{ow}$.

Authors should explain further in addition to Line 208-210 page 11, why F6-hGO possesses the highest hydration capacity compared to other membranes as depicted in Figure 3(e) F6-water amount in the hydration layer of the proposed surfaces?

Reply:

Thank the reviewer for the valuable guidance. The medium-length perfluoroalkyl chains (C_6) exhibit the highest hydration capacity as the surface possesses more uniform orientation of the interfacial water compared with others. The uniformity in hydration

structure offers more available water interaction sites, affording larger amount of interfacial water and higher hydration capacity. We analyzed and discussed the reason why F₆-hGO possessed the highest hydration capacity in the next paragraph. We have added the corresponding discussion in our revised manuscript (Manuscript Page 9 Line 207-209):

“We also observed a change in the water amount in the hydration layer, that is, the value first increases and then decreases with the perfluoroalkyl chain length (Figure 3e). The variation in hydration capacity is considered to be related to the interfacial hydration structure induced by the perfluoroalkyl chains with different lengths, which will be discussed in detail later. ... We defined α as the angle between \vec{u} (the opposite vector of water dipole) and \vec{z} (the normal vector of the surface) to estimate the spatial orientation of interfacial water to explore the underlying molecular-level details for the variation in the water amount of the hydration layer. We divide the hydration layers into two regions and compare the probability distributions of $\cos\alpha$ in the two regions (Figure 3f). It can be seen that most of the water bears the distributions of $\cos\alpha$ centered ~ -1 , implying water mainly favor orienting with two O–H bonds directed downwards towards the surface in our models. With the presence of perfluoroalkyl chains, the orientation of the water molecules is perturbed due to the electrostatically dominated interactions between the water and the perfluoroalkyl chains⁵². This electrostatic interaction is sensitive to chain length and position, which was adequately discussed in detail in the previous work⁵². In our study, the water orientation distribution in the regions I and II is more similar in the case of F₆-hGO, which implies a more uniform hydration structure on the surface. The uniformity in hydration structure offers more available water interaction sites and thus reaches a larger amount of interfacial water and higher hydration capacity⁵⁰. This nonlinear dependence of the hydration capacity to the perfluoroalkyl chain length reveals that an appropriate chain length exists for the similar amphiphilic surfaces to exhibit the strongest hydration capacity.”

Authors should place all figures within or next to the text of the subsection, not above the text or paragraph.

Reply:

Thank the reviewer for the guidance. We have put all figures next to the text of the subsection.

Line 284 page 14, authors should be presented as “3 Conclusion” instead of “3 Discussion”.

Reply:

Thank the reviewer for the guidance. We have presented the title of subsection 3 as “3 Conclusion”.

Reviewer #3 (Remarks to the Author):

This is an interesting article about self-assembled amphiphilic structures on membranes. While some properties like the contact angle rise as expected with increasing alkyl chain length, other properties like the oil adhesion force show a more differentiated picture. Obviously, hydration and morphology are more relevant than pure wettability. The work is well backed up by state-of-the-art spectroscopy. The interpretation of the data is supported by simulations – but I am unable to judge their quality. What is missing is the question how the coatings deal with scaling – I recommend additional experiments to add this characterization. I find the publication interesting, and I am in favor of its publication after major revision.

Thank the reviewer for the highly positive remarks and valuable guidance.

As to the question how the coatings deal with scaling, the gypsum scaling experiment was performed. The hexadecane-in-water emulsion with 20 mM sodium sulfate and 20 mM calcium chloride was used as the feed solution to conduct the filtration. After treating the emulsion for 6 h, no significant flux decline was observed as presented in Figure S12. The surface morphology of the membrane before and after the scaling experiment was tested by SEM. As shown in the insert of Figure S12, no gypsum scaling can be found on the membrane surface.

We have added the discussion in our revised manuscript and Supplementary Information (Manuscript Page 12 Line 264-266):

Manuscript: In addition, the membrane exhibits a comparable pure water flux as the initial one after multiple cycles of fouling with oils and cleaning with water (Figure 4e). For the challenge of scaling, no significant flux decline was observed after operating the gypsum scaling experiment for 6 h (Supplementary Figure S12). Besides, the F₆-hGO membrane shows a promising opportunity in the antifouling treatment of various emulsions with high rejection (>98%, Supplementary Figure S13) and low DR_i (< 10%, Figure 4f).

Figure S12 Time-dependent normalized flux of the F₆-hGO membranes during scaling experiment. The insert is the surface morphology of the F₆-hGO membrane before and after the scaling experiment.

The gypsum scaling experiment was performed. The hexadecane-in-water emulsion with 20 mM sodium sulfate and 20 mM calcium chloride was used as the feed solution to conduct the filtration. After treating the emulsion for 6 h, no significant flux decline was observed as presented in Figure S12. The surface morphology of the membrane before and after the scaling experiment was tested by SEM. As shown in the insert of Figure S12, no gypsum scaling can be found on the membrane surface.

Detailed comments

In the introduction on amphiphilic coatings as marine antifouling coatings, the list of references seems incomplete and the important contributions of groups like the Ober group (Cornell), the Webster group (NDSU), and the Laschewsky group (Potsdam) are not mentioned. This should be extended. Also, the work of the Ulbricht group (University of Duisburg-Essen) about amphiphilic coatings in membrane science is not cited. This overview needs to be complete.

Reply:

Thank the reviewer for the valuable guidance. We have extended our introduction

according to the work of the above-mentioned research groups (Ref. 18~22, 26, 28, 29).

In the introduction, the state of the art for the used assembly strategy using PA and PFCA on GO is missing and needs to be added.

Reply:

Thank the reviewer for the valuable guidance. For the construction of PA on membrane surfaces, assembly strategies based on hydrogen bonds and coordination are recently proposed. Coordination-driven metal-bridging assembly, in our study, is employed to perform the controllable assembly of PA and perfluoroalkyl chains. We have added the above introduction in our revised manuscript (Manuscript Page 4 Line 77-80):

“We sequentially assembled hydrophilic phytic acid (PA) and hydrophobic perfluorocarboxylic acids on a GO membrane to form an amphiphilic surface with a continuous hydrophilic domain and discrete hydrophobic domains. For the construction of PA on membrane surfaces, assembly strategies based on hydrogen bonds and coordination are recently proposed^{37, 38, 39}. Coordination-driven metal-bridging assembly, in our study, is employed to perform the controllable assembly of PA and perfluoroalkyl chains.”

While the selection of perfluorocarboxylic acid is discussed in detail, the choice of phytic acid is not explained. The authors should add why the specific compound has been selected.

Reply:

Thank the reviewer for the valuable guidance. In terms of the construction of antifouling amphiphilic surfaces, high hydration capacity is favorable for providing steric and energetic barrier to resist pollutants. Phytic acid (PA) possesses six phosphate groups with hydration energy of 44.4 kJ mol⁻¹ (*Phys. Chem. Chem. Phys.*, 2006, 8, 4530–4542), showing superior hydration capacity. So, we selected PA as the hydrophilic compound of our amphiphilic surface. We have added the above description in our revised manuscript (Manuscript Page 4 Line 91-94):

“PA possessing six phosphate groups with high hydration energy and

perfluorocarboxylic acid with ultralow surface energy were selected as the hydrophilic and hydrophobic components, respectively, and the amphiphilic surfaces were prepared on GO membranes via coordination-driven metal-bridging assembly. We chose perfluorocarboxylic acids with different numbers of carbon atoms, namely, 4, 6, 8, and 10, to adjust the length of hydrophobic chains and changed the concentration of Fe^{3+} on the surface to control the density of the perfluorocarboxylic acids, hence achieving the molecular-scale regulation of the surface chemistry.”

In the XPS analysis on page 6, line 115, the F/P ratio was determined to understand the stoichiometry of the two compounds on the surface. As the assembly process is a subsequent buildup on the surface, I am wondering if the F is predominantly present at the surface and the P might be buried? The authors have to briefly argue in the article if the quantification would also be valid if there is a layered system and which deviations could occur, accounting for the high surface sensitivity of the used method.

Reply:

Thank the reviewer for the valuable guidance. Perfluoroalkyl chains are predominantly presented at the surface due to the subsequent buildup. But the perfluoroalkyl chains would not completely cover the PA layer as the perfluoroalkyl chains are discretely assembled on the surface at a molecular scale. Besides, in our study, the change in the length of perfluorocarboxylic acids was less than 1 nm, which would not significantly affect the investigation of P element of XPS with the detection depth of 3~10 nm. The peak area of P element shows negligible change (Supplementary Figure S3), confirming the above statement. So, we believe that the quantification of the F/P ratio to determine the stoichiometry of the surfaces is valid. We have added the above discussion in our revised manuscript (Manuscript Page 5 Line 111-113):

“The variation in the length of perfluorocarboxylic acids is supposed to exert a negligible effect on the coordination process, which can be proved quantitatively by the area ratio of F peak to P peak in the XPS spectra of the F-hGO membranes. The peak areas of P element are almost consistent in the four cases (Supplementary Figure S3a), suggesting the detection of P element is not significantly affected by the chain length,

which illustrates the validity of the F/P ratio to characterize the F content. As shown in Figure 1d, the F content continuously ascends with the chain length, and its variation is almost in line with the molar ratio of the F atoms in the introduced perfluorocarboxylic acids, implying the distribution density of the coordinated perfluorocarboxylic acids on the four surfaces is almost similar.”

The discussion section is mainly a summary – here an extended discussion based on the literature state of the art is required with the corresponding references. Compared to which systems is the developed system superior and how does it compare to the current gold standard?

Reply:

Thank the reviewer for the valuable guidance. We have presented the title of subsection 3 as “3 Conclusion” to match our content. We elaborately discussed the advance of our membrane based on the corresponding references in subsection 2. Compared with the currently reported GO-based membranes and polymeric membranes for oil/water separation, the F₆-hGO membrane shows outstanding antifouling performance (Figure 4c, Supplementary Tables S2 and S3). More importantly, the as-prepared membranes can maintain their ultralow fouling tendency even at a relatively high initial flux of ~310 L m⁻² h⁻¹ (i.e. 620 L m⁻² h⁻¹ bar⁻¹, Figure 4d), which confirms their robustness in resisting oil fouling and advantage over the state-of-the-art membranes. The introduction and optimization of multiple antifouling mechanisms on the membrane surface account for the excellent antifouling properties. The hydration barrier and steric repulsion of the hydrophilic PA coating resist the spread of oil droplets, making the oil contact the membrane surface with a small area. The perfluoroalkyl chains with low surface energy endow the membrane surface with nonstick properties and fouling release behavior. Most importantly, in our case, the interfacial interactions of the proposed amphiphilic surface are regulated by the perfluoroalkyl chain length. There is an optimal chain length, namely C₆, to endow the surfaces with the lowest oil adhesion. At this length, the interfacial water is oriented more uniformly, thereby forming a hydration layer with more water. This enables the surface to maintain the hydration

capacity when introducing the perfluoroalkyl chains, thereby synergistically optimizing fouling resistance and fouling release properties. Consequently, the F₆-hGO membrane harvests extremely low adhesion with oil droplets, allowing oil droplets to easily peel from the membrane surface instead of forming a filter cake layer, and thus resulting in the enhanced antifouling efficacy.

In the materials and methods part in the supplements, literature references are missing. While the theory part is well referenced, for the preparation of the coatings and for the separation and antifouling performance evaluation such reference to previous work is entirely missing.

Reply:

Thank the reviewer for the valuable guidance. We have supplemented corresponding references in the Materials and Methods part including the preparation of the coatings, the calculation of surface energy, and the evaluation of antifouling performances (Manuscript Ref. 37, 38, Supplementary Ref. 1-3).

REVIEWERS' COMMENTS

Reviewer #2 (Remarks to the Author):

The authors have adequately responded to my comments. The revised manuscript is well written and organised. I hereby recommend that the manuscript can be accepted in this form.

Best regards,
Dr. Yusuf Olabode Raji

Reviewer #3 (Remarks to the Author):

The authors revised the manuscript very well and answered most of my concerns. Though the discussion of the data in the literature context still falls a bit short I recommend publication of the article.

Response to reviewer's comments

Reviewer #2 (Remarks to the Author):

The authors have adequately responded to my comments. The revised manuscript is well written and organised. I hereby recommend that the manuscript can be accepted in this form.

Best regards,

Dr. Yusuf Olabode Raji

Thank the reviewer for the highly positive remarks.

Reviewer #3 (Remarks to the Author):

The authors revised the manuscript very well and answered most of my concerns. Though the discussion of the data in the literature context still falls a bit short I recommend publication of the article.

Reply:

Thank the reviewer for the guidance. We have added some discussion of the data in the literature context.

Compared with the currently reported GO-based membranes and polymeric membranes for oil/water separation, the F₆-hGO membrane shows superior antifouling performance (Fig. 4c and 4d, Supplementary Tables 2 and 3). DR_t of our membrane is far lower than that of the reported membranes with the index generally of 40~60%. More importantly, as shown in Fig. 4d, most membranes achieve ultralow DR_t at low initial fluxes. While the F₆-hGO membrane can maintain its ultralow fouling tendency even at a relatively high initial flux of $\sim 310 \text{ L m}^{-2} \text{ h}^{-1}$. This value exceeds most oil-water separation membranes, which confirms their robustness in resisting oil fouling and advantage over the state-of-the-art membranes.